# Assessment of Home-Based and Mobility-Based Exposure to Black Carbon in an Urban Environment: A Pilot Study

**DOI:** 10.3390/ijerph18095028

**Published:** 2021-05-10

**Authors:** Max Gerrit Adam, Phuong Thi Minh Tran, David Kok Wai Cheong, Sitaraman Chandra Sekhar, Kwok Wai Tham, Rajasekhar Balasubramanian

**Affiliations:** 1Department of Civil and Environmental Engineering, National University of Singapore, Singapore 117576, Singapore; ceeamg@nus.edu.sg (M.G.A.); e0338248@u.nus.edu (P.T.M.T.); 2Faculty of Environment, The University of Danang—University of Science and Technology, 54 Nguyen Luong Bang Street, Lien Chieu District, Danang City 50608, Vietnam; 3Department of Building, School of Design and Environment, National University of Singapore, Singapore 117566, Singapore; bdgckw@nus.edu.sg (D.K.W.C.); bdgscs@nus.edu.sg (S.C.S.); bdgtkw@nus.edu.sg (K.W.T.)

**Keywords:** black carbon, personal exposure, urban air quality, vehicular emissions

## Abstract

The combustion of fossil fuels is a significant source of particulate-bound black carbon (BC) in urban environments. The personal exposure (PE) of urban dwellers to BC and subsequent health impacts remain poorly understood due to a lack of observational data. In this study, we assessed and quantified the levels of PE to BC under two exposure scenarios (home-based and mobility-based exposure) in the city of Trivandrum in India. In the home-based scenario, the PE to BC was assessed in a naturally ventilated building over 24 h each day during the study period while in the mobility-based scenario, the PE to BC was monitored across diverse microenvironments (MEs) during the day using the same study protocol for consistency. Elevated BC concentrations were observed during the transport by motorcycle (26.23 ± 2.33 µg/m^3^) and car (17.49 ± 2.37 µg/m^3^). The BC concentrations observed in the MEs decreased in the following order: 16.58 ± 1.38 µg/m^3^ (temple), 13.78 ± 2.07 µg/m^3^ (restaurant), 11.44 ± 1.37 µg/m^3^ (bus stop), and 8.27 ± 1.88 µg/m^3^ (home); the standard deviations represent the temporal and spatial variations of BC concentrations. Overall, a relatively larger inhaled dose of BC in the range of 148.98–163.87 µg/day was observed for the mobility-based scenario compared to the home-based one (118.10–137.03 µg/day). This work highlights the importance of reducing PE to fossil fuel-related particulate emissions in cities for which BC is a good indicator. The study outcome could be used to formulate effective strategies to improve the urban air quality as well as public health.

## 1. Introduction

Exposure to black carbon (BC) is of health concern, particularly in developing countries, as it is a carrier of toxic pollutants such as polycyclic aromatic hydrocarbons (PAHs) [1,2,3]. Epidemiological studies have revealed that the short-term and long-term human exposure to BC can lead to cardiovascular [4,5] and respiratory [6,7] diseases and increased mortality [8,9]. Personal exposure (PE) to airborne particulate matter (PM) containing BC in microenvironments (MEs) depends on a number of factors, including age and gender, and the time spent, activity and ventilation conditions in the specific ME [10,11,12,13]. Understanding and quantifying PE patterns related to PM-bound BC provides useful inputs to the development and validation of exposure models [14,15]. BC is mainly found in fine particulate matter (PM_2.5_, aerodynamic diameter <2.5 µm). PM_2.5_ exposure levels far exceed the World Health Organization (WHO) Air Quality guideline of the annual average concentration of 10 µg/m^3^ within India [16]. The health impacts of BC alone are much higher compared to other chemical components in PM_2.5_ when the parameters are expressed in the same unit of µg m^−3^ [17]. As India is the second largest emitter of BC in the world [18,19,20], a systematic assessment of PE to BC merits a serious consideration.

Assessment of PE to BC using handheld, portable devices have received considerable attention [21,22,23,24,25]. Recent studies have indicated that among urban MEs, PE to BC occurs predominantly in transport MEs where the internal combustion of fossil fuels in on-road vehicles has been the major source of emissions of PM-bound BC. PE to BC has been investigated for pedestrians [26] and cyclists [27,28,29], and during several modes of transport by commuters, including motorcycles, passenger cars, taxis, buses, and trains [21,30,31,32,33,34,35]. PE to BC has also been characterized on commercial flights between two countries [36], within a country and airports [37]. In most of these studies, it was found that commuters’ PE to elevated levels of BC is of concern and may result in negative health outcomes. To provide better insights into the relationship between the PE to BC and potential health risk, studies have been carried out as a function of time-activity patterns over a combination of several MEs during daily activities, which indicated pronounced potential health impacts from transport MEs [12,22,38,39]. Domestic cooking activities can also increase the potential health risk due to the PE to BC emitted from cooking fuels, the type and intensity of cooking and food ingredients used [40,41,42,43,44].

Trivandrum, a tropical coastal city at the southern tip of the Indian peninsula with a population of about 1 million, was chosen for this study because of the following rationale. Population and economic growth in Trivandrum, the capital of the state of Kerala, have resulted in the increased volume of on-road vehicles, which resulted in the deterioration of ambient air quality [45,46]. However, the impact of this source of air pollution on PE to BC remains unknown. Several previous studies have established the characteristics and anthropogenic nature of BC in Trivandrum based on observational data obtained at an urban background air quality monitoring station, with a mean background BC concentration in the range of 1.50–6.77 µg/m^3^ [47,48]. Interestingly, during a 2-day lockdown with no vehicles operating in the city, the BC levels dropped to 22% of the average concentrations before the lockdown, highlighting the influence of on-road vehicular emissions on BC levels [46]. Previous studies related to the PE to BC in India were conducted inside a commonly used vehicle (autorickshaw, New Delhi) [49], during daily activities of local residents in New Delhi [50], in domestic kitchens in rural North India [40,51] and among the general population in peri-urban South India [52]. However, the inhaled dose of BC due to PE in diverse MEs was only considered in the study by [52], which mainly focused on predicting at-residence BC levels using exposure modelling and validating it against 24 h BC profiles. Certain members of the local society (e.g., homemakers, elderly citizens) may be spending most of their time at home (indoors) while other individuals who remain mobile may be exposed to various levels of BC in diverse outdoor and transport MEs apart from home microenvironments. Therefore, it is important to gain a better understanding of home-based versus mobility-based PE to BC in urban environments so that effective air quality management strategies can be formulated to improve ambient and indoor air quality (IAQ) and more importantly to protect public health [53]. Such PE studies across MEs over 24 h are seldom conducted in cities [54].

In this pilot study, we assessed and quantified the PE to BC in Trivandrum by addressing two scenarios: (i) home-based PE assessment for 24 h and (ii) mobility-based PE assessment across transport MEs during the day and home-based PE assessment at the nighttime. For the mobility-based PE assessment, the volunteer visited different places of interest in the city (temples, shopping areas, restaurants, and friends and relatives’ homes) using two different modes of transport (a motorcycle and a private car). Inhaled dose values were estimated for BC in each ME, and the cumulative contribution of each ME to the daily integrated inhaled dose of BC was quantified. This mode of PE assessment facilitated the identification of BC hotspots in the urban environment for the home-based and the mobile participants. To our knowledge, the research reported here constitutes the first study of its kind, focused on the PE to BC of mobile city commuters and non-mobile individuals in residential homes.

## 2. Materials and Methods

### 2.1. Site Description and Experimental Design

Home-based PE measurements were conducted from December 2018 to January 2019 in a stand-alone house similar to other houses in the city, located in a residential area with no industrial activity in the vicinity on the western part of Trivandrum (Thiruvananthapuram, 8.52° N, 76.93° E, 3 m asl) situated near the south-west coast of India (see Figure 1). The house is situated in the city center about 90 m from a north-south road, which leads to a major hospital to the northern side and the airport to the south. The Arabian Sea coast is about 2 km east of the house. The prevailing meteorological conditions during the study were influenced by north-easterly winds and dry conditions which are typical for this time of the year. No unusual meteorological events were observed during the study period. Trivandrum, due to its coastal location, experiences land/sea breeze, resulting in a diurnal change of the wind direction and speed.

The BC measurement protocol involved the use of two portable BC monitors (described in detail in Section 2.2) by the two participants (volunteers, non-smoking, male, aged 30–40 years old), with one participant continuously monitoring the PE to BC in the indoor environment with high temporal resolution (1 min) and the second participant doing the mobility-based PE assessment in both outdoor and indoor environments (1 min). PE measurements of BC at the residential home (two-storey building) were made at the ground floor level by placing the monitor in the living room (between 0800 and 2200 local time) at the breathing zone height (1.5 m) and the bedroom (2200-0800 LT) at the height of the bed (0.7 m) where the participants were sleeping. The BC exposure concentration in the residential home was relatively uniform. Therefore, the living room location is representative of the PE to BC. Mobility-based PE measurements were carried out by placing the BC monitor inside a backpack with the sampling inlet protruding from the backpack in close proximity to the breathing zone. Whenever the second participant ventured to any location outside of the home environment, the second BC monitor was placed in the backpack with adequate caution given to fixing the inlet such that it would remain in the same position. Also, during mobility-based PE measurements, periodic backpack inspections were made to ensure that the BC monitor was operational, and the sampling tube was not dislodged. Our PE to BC study was conducted for a duration of 10 days for each participant.

### 2.2. Black Carbon Measurements

BC data were acquired continuously using a portable BC monitor (microAeth AE51; AethLabs Inc., San Francisco, CA, USA). The microAeth measures light absorption at 880 nm on a 3 mm diameter sample spot (I) and an aerosol-free reference spot (I_0_) in a T60 Teflon-coated glass fiber filter media to obtain attenuation coefficients at the wavelength of 880 nm (referred to hereafter as σ_AE51, 880 nm_) according to:(1)$$\sigma_{\mathrm{AE}51,\;880\;\mathrm{nm}}=\frac{A\;\times \;\Delta \mathrm{ATN}}{100\times \;F\;\times \;\Delta t}$$
A: area of the aerosol collecting spot (0.071 cm^2^), F: flow rate (150 mL/min), ATN: light attenuation calculated from 100 × ln (I_0_/I), ΔATN: change of ATN during the sampling interval Δt (60 s).

The mass concentration of BC is converted directly from $\sigma_{\mathrm{AE}51,\;880\;\mathrm{nm}}$ using the mass-specific cross-section of BC ($\alpha_{abs}=$12.5 m^2^/g, provided by the manufacturer) (Equation (2)).
(2)$$BC_{o}=\frac{\sigma_{\mathrm{AE}51,\;880\;\mathrm{nm}}}{\alpha_{abs}}$$

To maintain BC data integrity, several corrective procedures were applied to the collected data. In order to limit a bias from the filter loading of BC on the filter substrate, raw data with attenuation (ATN) values above 80 were not used in the analysis as recommended by [55]. The mass concentration of BC measured using AE51 can be underestimated due to BC loading of the filter, which alters the absorption and scattering of light. Hence, the BC data were corrected for filter loading effects using the procedure proposed by Kirchstetter and Novakov [56] as shown in Equation (3):(3)$$\mathrm{BC}\;={\mathrm{BC}}_{o}\;{\left(0.88\mathrm{Tr}\;+0.12\right)}^{-1}$$
BC: corrected black carbon concentration, BC_o_: instrument-reported concentration, Tr = exp(ATN/100): microAeth filter transmission that is calculated from the instrument-reported attenuation coefficient (ATN).

We applied Equation (3) to all BC measurements. The measurement precision of the microAeth is ±0.1 μg/m^3^ at 1-min average and a 150 mL/min flow rate with a measurement resolution of 0.001 μg/m^3^. For quality control, data processing, and analysis, the reader is referred to the Appendix A.

### 2.3. Fixed Site PM Measurements and Meteorological Parameters

Hourly averaged PM_2.5_ and meteorological (temperature, relative humidity, wind direction, and wind speed) measurements were acquired from the Central Pollution Control Board website (https://app.cpcbccr.com/ccr/#/caaqm-dashboard-all/caaqm-landing, accessed on 15 March 2020) for Trivandrum for the study period. The location of the automated air quality measurement site is shown in Figure 1.

### 2.4. Personal Exposure Assessment

The PE of an individual to air pollutants is a function of the concentration (C) and time (t) spent in a particular ME [57]. The exposure concentration to BC (*E_BC_,_h_* in µg/m^3^) of the participant staying at home (*C_home_*) was calculated as the indoor BC concentration averaged over the time period that the individual is home-bound (*t_home_*), shown in Equation (4). Similarly, the calculation of the temporally weighted aggregation of BC exposure in the range of MEs encountered of the mobile participant (*E_BC_,_m_* in µg/m^3^) is shown in Equation (5).
(4)$$E_{BC,h}=\frac{C_{home}\times t_{home}}{t_{home}}$$
(5)$$E_{BC,m}=\frac{C_{home}\times t_{home}\;+\;\sum C_{i}\times t_{i}}{t_{home\;}+\;\sum t_{i}}\;\;\;$$
*t_home_*: duration (h) that the participant had spent in the residential home, *t_i_*: duration (h) the participant had spent in other MEs (transport, restaurant, temple, leisure AC, leisure non-AC, bus stop).

### 2.5. Inhaled Dose Calculation

The inhaled dose was estimated based on the exposure concentrations (µg/m^3^) obtained in each microenvironment over the time spent (i.e., hour/day) in the corresponding ME and the inhaled rate (m^3^/hour). Calculation of the inhaled dose of BC was done according to Equations (6) and (7):(6)$$\mathrm{Integrated}\;\mathrm{inhaled}\;\mathrm{dose}=\overset{n}{\underset{i=1}{\sum }}\left({\mathrm{Concentration}}_{i}\times {\;\mathrm{Inhalation}\;\mathrm{rate}}_{i}\times {\;\mathrm{Exposure}\;\mathrm{time}}_{i}\;\right)$$
(7)Inhalation rate =Tidal volume×Breath frequency

The participants were not involved in intense physical activities such as running and cycling during the study period. The tidal volume chosen was 750 cm^3^ per breath and the typical breathing frequency selected was 0.20 breaths per second for all-day activities of male adults [58].

## 3. Results

### 3.1. BC PE Concentrations at the Home Microenvironment and During Urban Mobility

Figure 2 shows the time series of 48-h PE to BC from the data measured at the residential home and during urban mobile activities. It can be observed that for short periods of time the BC values, while commuting by a car and a motorcycle, exceeded 100 µg/m^3^ whereas the BC exposure experienced at home was significantly lower, i.e., within the range of 5–20 µg/m^3^. There were no major indoor activities other than cooking during the course of our IAQ measurements. BC emissions from cooking were likely to be low as the boiling of food items only took place using an LPG gas stove. During the motorcycle ride, the participant was directly exposed to vehicular emissions, whereas during the car commute the air-conditioning system was in operation. This resulted in greater variability of BC exposure for the mobile participant when riding the motorcycle whereas the BC exposure in the car was fairly consistent, as can be seen in Figure 3. As also shown in Table 1, the lowest BC concentration was observed at the home ME with a geometric mean (GM) of 8.27 ± 1.88 µg/m^3^. In contrast, travel by the motorcycle and the car exhibited the highest PE concentrations with GMs of 26.23 ± 2.33 µg/m^3^ and 17.49 ± 2.37 µg/m^3^, respectively with motorcycle exposure being significantly higher (*p*-value < 0.001, non-parametric Mann-Whitney test) than inside car. Temple (GM: 16.58 ± 1.38 µg/m^3^), naturally ventilated restaurant (GM: 13.78 ± 2.07 µg/m^3^), bus stop (GM: 11.44 ± 1.37 µg/m^3^), and leisure activities in naturally ventilated (GM: 9.04 ± 1.92 µg/m^3^) and air-conditioned MEs (GM: 7.92 ± 1.55 µg/m^3^) showed BC concentrations that were still higher than the ones at the residential home. The maximum BC concentration values recorded, while commuting, further underscore the significant short-term PE to BC with concentrations of 381.62 and 288.18 µg/m^3^ during the ride by motorcycle and car, respectively. Table A1 lists the temporally weighted aggregation of the BC PE showing that the mobile participant on each day and on average (11.99 µg/m^3^) was exposed to higher BC concentrations than the home-based participant (10.20 µg/m^3^). The means of BC concentrations measured at home at different times of the day (morning, noon, afternoon, evening, and night), shown in Table A2 were significantly different (*p*-value < 0.001, non-parametric Kruskal–Wallis test), and showed a diurnal pattern with maxima in the morning (06:00–11:00), evening (17:00–21:00), and night (21:00–06:00) and minima during the day.

Interestingly, the variability of PM_2.5_ levels (see Figure 2) monitored at the fixed monitoring station, which is about 2.1 km away from the residential home, was reasonably consistent with the BC levels measured at the home ME. Figure A3 shows the correlation between PE to BC at the diverse MEs and the PM_2.5_ values recorded at the fixed monitoring station. The correlation was moderate to strong in the residential home (r = 0.58), leisure natural ventilation (natural ventilation termed NV hereafter) (0.62), restaurant (0.78), and temple (0.86) MEs while it was the lowest for leisure AC (air conditioning termed AC hereafter) (0.07), car (0.07), and motorcycle (0.23).

### 3.2. Daily Integrated BC Dose on Five Different Days

Figure 4 shows BC inhaled dose values for five representative days on which simultaneous mobility and home-based BC PE observations were recorded. For the mobility-based PE assessment, the participant was mobile, i.e., spending time in the city commuting to different MEs such as restaurants, or pursuing leisure activities in non-air-conditioned and air-conditioned spaces, whereas the home-based PE assessment refers to the PE to BC experienced by the other participant while remaining at home with no participation in outdoor activities (on the same five days for consistency). The corresponding BC values are listed in Table A3. In general, as can be seen in Figure 4, the daily inhaled dose of BC was higher for the participant involved in outdoor activities compared to the participant remaining at home. The differences in the inhaled dose values between each day ranged from 29% for day 1 to 8% for day 3. The mobile participant spent 68–80% of 24 h in the home environment. The remaining time was spent in transport MEs (5–12%) and other MEs (12–20%). The secondary major contributor to the total daily BC inhaled dose was related to local transport on most of the days (21.1% on average) with the exception on day 3 where a restaurant visit contributed 22% to the total daily intake and transport only 8%.

### 3.3. Indoor to Outdoor Ratios of BC PE

The simultaneous home-based and mobility-based BC PE observations allowed us to calculate the mean home-based (HB) to the mobility-based (MB) BC PE ratios, HB/MB (see Table 1). Overall, the HB/MB BC ratios of the mobility-based participant were less than 1 in all MEs except for the bus stop (1.45) and leisure NV (1.11). For motorcycle (0.58) and car (0.69), and temple (0.74), the HB/MB ratios were fairly low while they were close to 1 for the restaurant (0.93) and leisure AC (0.96).

## 4. Discussion

The integrated PE to BC over 24 h and its health outcomes are determined by air quality levels prevailing in indoor as well as outdoor MEs. Since urban dwellers tend to spend up to 90% of their time in indoor environments [59], it is of particular importance to consider the IAQ’s contribution to PE on a day-to-day basis. Our results for the home-based participant (and the mobile participant when at the same home) show that the BC exposure at the residential location (8.27 µg/m^3^) was elevated by a factor of approximately 2–3 when compared to the fixed site monitoring BC data reported in the literature for Trivandrum [45,47]. Also, compared to a PE study by [50] involving observations of BC at a home environment in New Delhi (2.78 µg/m^3^), our observations are significantly higher. In fact, the mean BC concentration, a minor component of PM_2.5_, in the residential indoor environment is even similar to the WHO’s annual air quality guideline value of 10 µg/m^3^ for PM_2.5_.

There are several factors that influence IAQ including measurement locations, indoor activities, the mode of ventilation (e.g., natural ventilation or air conditioning), and building design [11,60,61,62,63,64]. In recent years, air quality studies have focused on urbanized areas where an increasingly large proportion of the world’s population lives (55%), but is also exposed to traffic-related pollution due to the high number of vehicles and the proximity of homes, offices, and leisure places to roads as well as PM emissions from haze episodes [60,61,65,66,67].

With respect to India, BC levels inside homes have been shown to be enhanced due to the use of household fuels such as wood and coal with less combustion efficiency [40,68]. However, for our study, the influence of cooking fuels on the measured BC concentrations appears to make a minor contribution as liquid petroleum gas (LPG) was used as cooking fuel which emits significantly less PM compared to wood, coal, and oil [69]. The residence of the home-based participant is approximately only 90 m away from a main road carrying a high traffic volume. We carried out concurrent measurements of BC over 48 h inside the home and at a location just outside of it to study the influence of outdoor air quality (OAQ) on IAQ. The indoor to outdoor BC ratio was nearly one (I/O BC ratio ~ 1), indicating that the outdoor BC influenced the BC levels indoors, when there were no major indoor air pollution sources. Therefore, the high PE BC concentrations can likely be attributed to vehicular emissions of PM_2.5_. This interpretation is consistent with the fact that ambient air quality in urban areas is often substantially influenced by emissions from vehicular traffic, particularly in South and Southeast Asia [70,71,72,73,74].

For the mobile participant in our study, the BC exposure is a combination of BC concentrations encountered in several MEs. The participant’s highest BC exposure was observed during the travel by the motorcycle (26.23 µg/m^3^) and car (17.49 µg/m^3^), suggesting that the health risk associated with urban mobility during peak commuting hours is significant. Our observations pointing to PE to high BC levels during commuting in India are consistent with the findings from BC PE studies reported by [49,50]. Autorickshaws (23.40 µg/m^3^) and buses (14.10 µg/m^3^) were reported to be major BC emission sources by [50]. However, autorickshaws (42.00 µg/m^3^) were the only mode of transport investigated by [49]. The BC PE levels inside the car were approximately 33% lower compared to the motorcycle for the mobile participant which can be attributed to the operation of the AC system (used in a recirculation setting mode with windows closed). The filtration system of the AC prevents the entry of airborne particles into the in-cabin environment, thus reducing the PE to BC concentrations. The association between ventilation settings and in-vehicle exposure has been reported with a decrease in air pollutant concentrations when windows were closed and the AC was on [75,76,77]. However, during motorcycle trips, no PM mitigating device was in place, which exposed the rider directly to freshly emitted airborne particles from the fossil fuel combustion in other on-road vehicles (motorcycles, cars, bus, and trucks), accounting for the higher BC concentrations [49,77].

For the mobile participant, we observed high BC concentrations while carrying out daily routine tasks. We categorized these MEs as restaurants, leisure places equipped with AC (supermarket, photostudio, bakery, and polyclinics) and naturally ventilated leisure places (community centers and fruit stores). Furthermore, PE to BC was investigated at bus stops in the city. All of the aforementioned MEs are locations in urban areas, which the local population frequently visits during their daily life, and were hence chosen as representative areas to determine the BC PE levels. The BC PE concentrations at all the MEs are higher than at the home environment (8.27 µg/m^3^), with the exception of leisure place with AC (7.92 µg/m^3^), which can be attributed to the filtration of PM by the AC systems in place at the respective buildings [78]. High PE to BC levels was experienced at the temple (16.58 µg/m^3^), likely due to the burning of incense sticks and candles, which are accompanied by high emissions of PM and BC [74,79,80,81]. Elevated BC PE concentrations were also observed at bus stops (11.44 µg/m^3^). Bus stops are associated with intense and highly localized emissions of BC owing to various vehicular activities such as deceleration, idling and acceleration which have been shown to lead to higher BC emissions [74,82]. The high BC PE concentrations at restaurants (naturally ventilated, 13.78 µg/m^3^) are likely a result of the combination of ambient (outdoor) BC concentrations, cooking activities, and cooking fuels used [41,83].

The measurement of the indoor to outdoor (I/O) ratio of air pollutants, monitored at indoor and outdoor locations of the same building, plays an important role in identifying the potential source of air pollutants. This ratio facilitates the interpretation of factors that influence the migration of air pollutants between indoor and outdoor environments such as ventilation type (natural or mechanical) and infiltration. In our study, the MEs visited by the mobility-based participant were farther away from the home ME. Therefore, we have calculated the HB/MB ratio and used it in place of the I/O ratio. The low HB/MB ratios for BC further underscore that the mobility-based participant in most MEs was subjected to increased BC exposure in MEs in the city. The variation in the HB/MB values (0.58–1.45) is likely to be driven by outdoor concentrations of BC in a majority of the MEs for our study as indoor emission of BC from cooking was only observed at the restaurant while incense burning took place at the temple; other instances of potential BC emissions from cooking, smoking and/or incense burning in any of the other MEs or at the naturally ventilated home did not take place. Nevertheless, it should be noted that the filtration of outdoor air in the air conditioning system of the car (in-cabin) and leisure ME (indoors) tends to reduce the BC exposure. In our study, the HB/MB ratios estimated for BC are generally higher compared to I/O values calculated for BC in California, USA (0.6–0.65) by [84] and are comparable to those in Vietnam (0.96–1.09) [66], suggesting that the home ME in Trivandrum is strongly influenced by BC emissions in the city.

Previous studies on BC emissions in Trivandrum reported the influence of boundary layer dynamics and transport on BC concentrations [46,47]. Source apportionment results indicated that around 66% of BC originates from the local traffic [85]. The results from the mobile-based exposure scenario indicate high PE concentrations of BC during urban mobility (car and motorcycle). This observation reinforces the earlier findings from home-based PE observations that the main source of BC in Trivandrum is from local traffic-related activities. In addition, the moderate to strong correlation between BC and PM_2.5_ in several of the naturally ventilated MEs (i.e., home, leisure, restaurant, and temple) is indicative of the significant contribution of fossil fuel-based combustion emissions to PM in Trivandrum. The low correlation in the transport MEs for car (0.07) and motorcycle (0.23) may be due to the BC measurements taking place in close proximity to the source of combustion emissions (thus high BC values as shown earlier) and low number of data points recorded in these MEs compared to the other locations. This would suggest the inadequacy of data collected at fixed monitoring stations for health risk estimations.

The inhaled dose value of BC for the home-based participant ranged from 118.10 to 137.03 µg/day while it was from 148.98 to 163.87 µg/day for the mobile participant. These BC inhaled dose values were significantly higher than the BC inhaled dose values reported in other studies, for example, adults in Belgium (14.10–77.70 µg/day) [21], children in South Korea (6.60–46.30 µg/day) [38], and children in Italy (39.20 µg/day) [43]. In addition to adverse health effects, the PE to elevated BC levels over a short duration or moderate BC levels over an extended time period may harm people’s cognitive performance [86,87,88].

In general, studies investigating human health effects of BC have mostly relied on ambient air quality measurements to assess the exposure of local populations to BC [1] which does not fully represent their actual exposure [11,74] as evident from the current study. Sharma and Balasubramanian (2019) investigated the influence of traffic-related PM emissions at a naturally ventilated residence, which is located in close proximity to a road, and the possibility of mitigating the inhalation of PM for the occupants of the apartment [71]. Their findings indicate that potential long-term health effects are associated with the inhalation of PM-bound toxic trace elements and that the use of a mitigation device such as a portable air cleaner (PAC) reduces indoor PM concentration levels by up to 74% yielding tangible health benefits at an affordable cost and with energy efficiency. The use of a fan filter unit in student dormitories [89] has also demonstrated the effective reduction of PM_10_, PM_2.5_, and PM_1_ particle concentrations by 80.9%, 80.4%, and 78.5%, respectively.

## 5. Limitations of the Study

We acknowledge that our pilot study has certain limitations, such as the small number of participants and limited spatio-temporal profiles of BC in the aforementioned MEs of Trivandrum. The BC data were only collected with two volunteers for a short duration of time (10 days) in one season. The reason for this limitation is that our BC measurements were conducted only on dry days, rain events occurred on all other days from December 2018 to January 2019. The BC data in the transport MEs were limited to only two modes of transport and only one ventilation scenario for car (AC) for a particular engine type. Future studies may need to address these shortcomings. Additionally, more extensive PE to BC studies should be conducted over a larger geographical area and longer time duration with participation of more volunteers to obtain a more robust dataset which can be used to quantify health risk due to exposure to BC as done by [40]. Nevertheless, the practical implications of our study as stated below provide the impetus for conducting more research on PE to BC in cities.

## 6. Practical Implications

Our findings show that the PE to BC in urban environments within India is likely to remain elevated at indoor and outdoor MEs because of the increased population of vehicles and traffic congestion with no major expansion of city infrastructure. Consequently, city commuters are likely to be subject to a high level of health risk due to inhalation of BC and other air pollutants of traffic origin. The health status of homemakers, the elderly, and children could be affected adversely while remaining indoors in naturally ventilated buildings with no PM mitigation devices due to the migration of freshly emitted airborne particles from the outdoor environment. In order to effectively improve air quality in Indian cities and public health, we suggest that PE measurements using mobile devices be conducted to complement a wide network of fixed ambient air quality monitoring stations on the city scale as demonstrated by [74,90]. The use of portable air cleaners, or fan filter units indoors is a cost-effective measure to mitigate the PE to air pollutants when needed [89,90,91]. A conceptual study in this direction, incorporating real-time air quality information based on model simulation and measurements, has recently been demonstrated in Hong Kong by [53]. The outcome of these studies can be used by citizens to make informed decisions to reduce their personal exposure to airborne pollutants. In the long-term, investments should be made to decarbonize the transportation sector with introduction of electrical mobility together with active modes of transport such as walking and cycling.

## 7. Conclusions

Urban dwellers are consistently exposed to high levels of BC emitted from anthropogenic activities, but little is known about their personal exposure to BC levels. While several studies in urban areas focus on PM_2.5_ and its relevance as a health indicator, BC is considered a more accurate metric to study health implications. The findings from this study revealed that the inhaled dose of BC is enhanced for city commuters resulting in potentially higher health risks compared to individuals staying at home because of intense human activities in diverse MEs and the combustion of fossil fuels in the transport sector. While air-conditioned spaces in the city (such as AC buildings) had a lower level of BC as compared to the naturally ventilated residential home, the reduction was found to be only minor. In the short-term, the feasibility of other mitigation options (e.g., portable air cleaners and fan filter units) for naturally ventilated buildings should be explored. These mitigation technologies are both affordable for citizens living in developing countries and sustainable when viewed from a long-term climate perspective. Overall, our study shows that decarbonization measures have to be put in place in Indian cities like Trivandrum to provide co-benefits of air quality improvement, climate change mitigation, and healthy living. As this work was carried out as a pilot study with only two participants, future studies should investigate PE with a larger number of subjects for longer periods of time. In addition, such studies should also involve a more comprehensive analysis of BC and co-emitted air pollutants from the combustion of fossil fuels to present a more accurate representation of long-term health effects.

## Figures and Tables

**Figure 1 ijerph-18-05028-f001:**
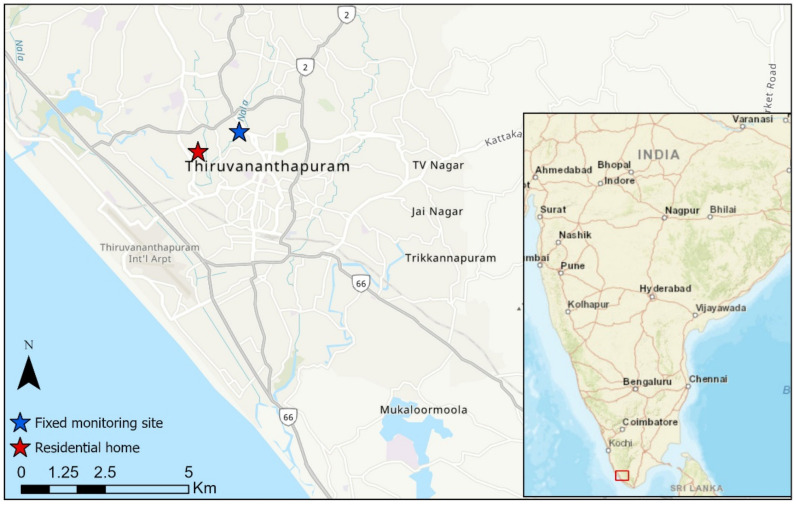
Map view highlighting Trivandrum; the red star highlights the residential home, whereas the blue star shows the location of the fixed monitoring site (2.1 km from the residential home), and (inset) overview of India.

**Figure 2 ijerph-18-05028-f002:**
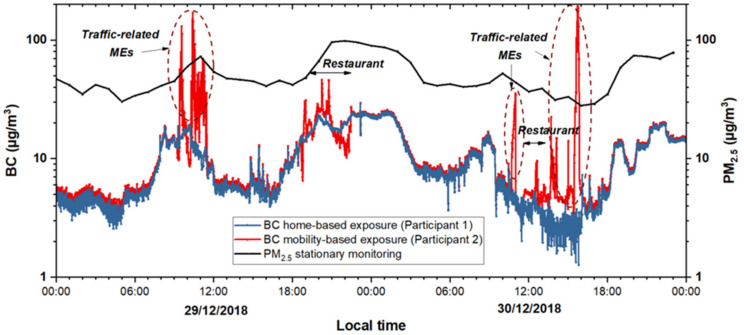
Time series of personal exposure to BC over 48 h for the home-based participant 1 (blue), and participant 2 during mobility-based exposure (red) and stationary PM_2.5_ (black) values.

**Figure 3 ijerph-18-05028-f003:**
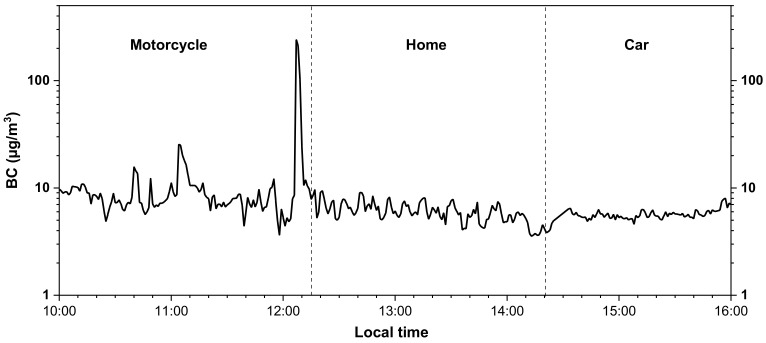
BC exposure for motorcycle, home, and car for the mobile participant.

**Figure 4 ijerph-18-05028-f004:**
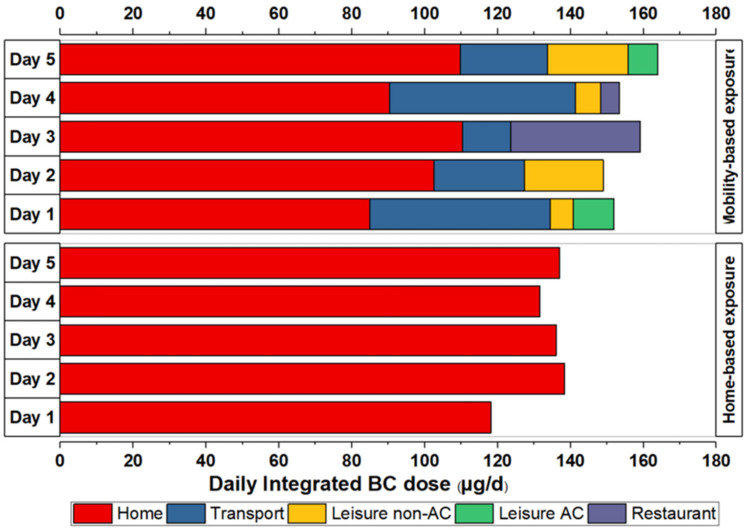
Daily integrated dose of BC during home-based exposure and during mobility-based exposure encompassing diverse MEs on representative days. Abbreviations: black carbon (BC), air conditioning (AC), microenvironments (MEs).

**Table 1 ijerph-18-05028-t001:** BC personal exposure concentrations and ratio of mean BC concentrations from home-based (HB) and mobility-based (MB) participants (simultaneous measurement, the ratios are calculated pairwise) in different MEs in Trivandrum.

Metric	Home	Transport				Restaurant (NV)		Leisure				Temple		Bus stop	
		Motorcycle		Car				AC		NV					
		Conc.	HB/MB ratio	Conc.	HB/MB ratio	Conc.	HB/MB ratio	Conc.	HB/MB ratio	Conc.	HB/MB ratio	Conc.	HB/MB ratio	Conc.	HB/MB ratio
AM	10.13	37.91	0.58	25.75	0.69	17.53	0.93	8.93	0.96	11.83	1.11	17.45	0.74	12.11	1.45
AM SD	8.82	41.11	2.34	30.72	0.85	11.60	0.27	5.98	2.77	12.49	0.60	5.83	0.54	5.08	0.34
GM	8.27	26.23	0.29	17.49	0.46	13.78	0.89	7.92	0.68	9.04	0.95	16.58	0.55	11.44	1.40
GM SD	1.88	2.33	2.60	2.37	2.71	2.07	1.33	1.55	1.79	1.92	1.86	1.38	2.19	1.37	1.36
Max	381.62	381.62	35.44	288.18	10.96	101.37	2.14	49.52	33.16	179.16	4.91	35.77	1.73	27.90	1.78
Min	0.00	3.08	0.02	0.52	0.01	3.63	0.31	1.16	0.12	2.61	0.07	9.55	0.20	7.55	0.60
CI	0.14	3.66	0.29	2.29	0.09	0.71	0.02	0.70	0.46	0.61	0.04	1.74	0.23	2.17	0.20

Leisure NV: community centers, vegetable store. Leisure AC: supermarket, photostudio, bakery, polyclinics. Abbreviations: home-based (HB), mobility-based (MB), arithmetic mean (AM), arithmetic mean standard deviation (AM SD), geometric mean (GM), geometric mean standard deviation (GM SD), confidence interval (CI).

## Data Availability

The data presented in this study are available on request from the corresponding author.

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
