# Peer review of "Assessment of Home-Based and Mobility-Based Exposure to Black Carbon in an Urban Environment: A Pilot Study"

_ijerph, 2021, doi:10.3390/ijerph18095028_

Round 1

Reviewer 1 Report

we had agreed that the manuscript is acceptable before. 

This manuscript is a resubmission of an earlier submission. The following is a list of the peer review reports and author responses from that submission.

Round 1

Reviewer 1 Report

This research proposes an interesting study to assess urban dwellers' personal exposure to black carbon, based on home-based and mobility-based scenarios, through a pilot study in the city of Trivandrum in India. This research fits the scope of the journal and is well written. 
However, from my point of view, the design of the experiments is not appropriate and should be improved.

1)Page 3 Line 123 - 128. This study involved two volunteers for assessing the pollution exposure. However, one participant only doing indoor and the other one doing both indoor and outdoor. Is there any reason for this design?

2) Page 3 Line 129 - 132. The authors claim the two monitors are used and placed two locations in the room. Then the authors only use living room sensors as representative. Then what is the reason for using two monitors? Moreover, the authors emphasize the cooking should be one of the factors for PE indoor. Why not replace the sensor in a place like the kitchen?

3) Page 5, section 2. 6 Inhaled dose calculation. The inhaled dose should be varying depending on the types of participants' activities (e.g., running, walking). The authors only consider average values for light activities.  (Maybe you should define what are the light activities here.)

4) In section 3.1, the authors compare the BC concentrations at fixed stations to the global station in table1.  However, this is obviously not a fair comparison. The temporal resolution(e.g. monthly, daily, hourly) of BC and the time are not clear. 

In my opinion,  the questions listed above should be addressed.  and the manuscript should be reconsidered after major revision 

Author Response

Pl see attached.

Reviewer 2 Report

This is an interesting study on BC which is innovative because usually we talk about PMs

there are some comments on the manuscript

-in the abstract it is not clear how diverse microenvironments 21 (MEs) during the day and in the same home were measured-how you simulated the microenvironemnts in the home? also I do not believe that the term microenvironment is correct here. it should be "other everyday situations" or something like this

-also I do not undestand how you state that eg the restaurant scenario is different to the bus stop scenario since many different restaurants and bus stops may exist in the city! it is probable that it is explained in the text but in the abstract it does not look plausible. please elaborate on this

-the manuscript needs proof reading by a fluent English speaker because there are some mistakes there. please do so in order to be able to have the paper published some examples are shwon here but there are many more

eg replace

Exposure to black carbon (BC) is of health concern, particularly in developing coun-34 tries, as it is a carrier of toxic pollutants such as polycyclic aromatic hydrocarbons

with

Exposure to black carbon (BC) is an important health concern, particularly in developing coun-34 tries, since this is a carrier of toxic pollutants such as polycyclic aromatic hydrocarbons

replace

can lead to cardiovascular diseases [4,5], respiratory [6,7] and increased 37 mortality [8,9]

with

can lead to cardiovascular  [4,5] or respiratory diseases [6,7] and increased 37 mortality [8,9]

you state

including the age, gender, 39 type of people in a specific ME

what do you mean by type of people??

-in case you use again a term eg PAH then you should use an abbreviation but if you do not use again the term there is no need for abbreviations

-lines 65-93 in introduction I do not like the extended bibliography you give on the reasons for this study. first of all you dont really give a reason for an international audience but on a local audience for the particular city. all the relevant discussion should be moved to the discussion section to compare your findings with others. If so many studies have been done for this city what is the novelty of your study?

-the introduction should be shortened in half

-in the materials and methods why this particular house was chosen? also you say that it is a stand-alone house-what does this mean? is it outside the city or it is in line with other houses? also how representative can one house be in hundreds of houses with different building ages, cleaning conditions etc? it looks quite random this way. please elaborate how this site was chosen and what the limiations are in the limitations part of the study

in 2.2 please do not give the explanations of the formula as a text because it is very tiring, give them as 

A: area of the aerosol collecting spot (0.071 cm2)

F: flow rate (150 147 mL/min) etc...

also I am sorry but I can nowhere see the explanation of σAE51,880nm?

the 2nd formulat how is it linked to the 1st? is BCo derived from the 1st formula?

please elaborate on all this

-please shorten 2.3 to only the necessary

-in 2.4 please give also the meteorological parameters website, not just the PM website

-the same in 2.5 and 2.6 please give the formulas explanation as asked before. again, how is ???,? etc used in the next formula Integrated inhaled dose. at least an abbreviation of your value eg IID is needed!

-I stronly object to Table 1. it makes no sense to compare with other studies in a Table! please use the table data to compare in a qualitative manner with your data

-I do not understand how the 2-person data can be compared in Fig 6 to be honest-how different in age, size, gender, smokimg habits etc were the people? has this been taken into account in the formula given? how representative is this scenario after all and for what population? please elaborate

-the lines 236-264 should be definitely shortened because it is very tiring for the reader, please keep only the most important data

-Fig 3, Fig 4 and Table 2 show the same data in different forms please keep only one

-The Fig 5 is interesting but you have described have you have done here in the materials and methods, either explain it or omit the figures. give more information on how representative the database was-how far from the subject was the PM value noted here? because if the PM was measured far from the subject then the correlation is not relevant! there is something wrong with the Temple Figure, why r is so high while p value is much higher than 0.05?

-discussion should be based on these findings and it should not be a general discussion on many more matters. please elaborate on the correlation results you found before. it should definitely be shortened to 2/3 of present length. please do not quote actual values that can be found in the tables and figures

-in limitations you state that were conducted only on dry days, rain events occurred on all other days from but I think in materials and methods you said that study were influenced by north-easterly 114 winds and dry conditions which are typical for this time of the year. No unusual meteor-115 ological events were observed during the study period. please elaborate on this

-it should be clearer in the discussion and in the limitations how representative this study is for a common resident of this city (and as such other similar mega cities around the world) so with what precautions this study should be used

-I believe that the chapter Practical Implications should be omitted unless it is necessary by the editor and all the important data should be shown in the conclusion or in the discussion

-please add the following references in relation to 

Papaoikonomou et al, Aerosol and Air Quality Research201818(6)pp. 1457–1469

Yadav et al Air Quality, Atmosphere and Health201912(7)pp. 775–783

Balmes Journal of Allergy and Clinical Immunology,
Volume 143, Issue 6,
2019,

Author Response

Pl see attached.

Round 2

Reviewer 1 Report

Thanks to the authors for the detailed response.  I think the comments are addressed properly. I recommend accepting this manuscript.

Author Response

Thank you for your constructive feedback.

We have proofread the revised manuscript thoroughly and improved its presentation style. The changes made in the text are indicated in red.

Reviewer 2 Report

The manuscript has been improved however there are still some limitations

-the standard deviation in the mean values as shown in the abstract is very high, please give some explanation for this otherwise it looks like bad design

-very little has been omitted from the manuscript. It was asked to considerably reduce a number of parts eg discussion, introduction so that the reader does not get tired, this did not happen. Please kindly refer to the previous review and act accordingly

-it was asked to include a number of updated references related to urban pollution, this did not happen. Please kindly refer to the previous review and act accordingly

Author Response

Thank you for your feedback. Our response is given below.

Comment 1: the standard deviation in the mean values as shown in the abstract is very high, please give some explanation for this otherwise it looks like bad design

Response: The standard deviation here reflects the temporal and spatial variations of BC (black carbon) concentrations measured during the field study. BC concentrations were measured in an uncontrolled environment. The standard deviation does not represent the reproducibility of BC data. This standard deviation is not to be mistaken with standard errors associated with measurements of experimental parameters in a controlled study. Given the scope of the study, the standard deviations reported are not high and such deviations are typical in air pollution measurements conducted in uncontrolled environments.

We have added the following text in the abstract to clarify this point:

The standard deviations represent the temporal and spatial variations of BC concentrations.

Comment 2: very little has been omitted from the manuscript. It was asked to considerably reduce a number of parts eg discussion, introduction so that the reader does not get tired, this did not happen. Please kindly refer to the previous review and act accordingly

Response: Since IJERPH is an interdisciplinary journal, we need to provide sufficient scientific background information on the topic addressed in the manuscript, point out knowledge gaps and indicate the novelty of our work. We wrote the introduction section accordingly. Reduction of the length of the manuscript will compromise its readability and more importantly the quality of our scientific outcomes as presented and discussed in the manuscript. This change might also affect the reputation of the journal. We, therefore, would like to retain the contents of the manuscript as they are now, which are appreciated by reviewer #1.

Comment 3: it was asked to include a number of updated references related to urban pollution, this did not happen. Please kindly refer to the previous review and act accordingly.

Response: The three suggested references are not related to our manuscript at all. Yet, we have cited one reference out of the three references.

We have proofread the manuscript thoroughly and improved its presentation style as well as its quality.